# Laser nanofabrication inside silicon with spatial beam modulation and anisotropic seeding

Rana Asgari Sabet [1,2], Aqiq Ishraq [2], Alperen Saltik [1], Mehmet Bütün [1] & Onur Tokel [1,2] ✉

Nanofabrication in silicon, arguably the most important material for modern technology, has been limited exclusively to its surface. Existing lithography methods cannot penetrate the wafer surface without altering it, whereas emerging laser-based subsurface or in-chip fabrication remains at greater than 1 µm resolution. In addition, available methods do not allow positioning or modulation with sub-micron precision deep inside the wafer. The fundamental difficulty of breaking these dimensional barriers is two-fold, *i.e.*, complex nonlinear effects inside the wafer and the inherent diffraction limit for laser light. Here, we overcome these challenges by exploiting spatially-modulated laser beams and anisotropic feedback from preformed subsurface structures, to establish controlled nanofabrication capability inside silicon. We demonstrate buried nanostructures of feature sizes down to 100 ± 20 nm, with subwavelength and multi-dimensional control; thereby improving the state-of-the-art by an order-of-magnitude. In order to showcase the emerging capabilities, we fabricate nanophotonics elements deep inside Si, exemplified by nanogratings with record diffraction efficiency and spectral control. The reported advance is an important step towards 3D nanophotonics systems, micro/nanofluidics, and 3D electronic-photonic integrated systems.

Silicon is unique for its use as the electronics industry substrate material and its prominent role in micro/nanophotonics[1–5]. The material has diverse applications in the near- and mid-infrared regime[6,7]. Nanolithography techniques enable unique Si platforms for studying new physics and exciting functionalities, including subwavelength and metamaterial technologies[3,6–9]. The plethora of nanoscale functionality is entirely limited to the wafer surface and thus may be identified as "on-chip"[3,10–12]. A distinct paradigm is to directly fabricate devices inside the bulk of Si, without altering the top or bottom surfaces of the wafer. The corresponding "in-chip" fabrication paradigm[13] is based on exploiting infrared lasers at wavelengths where the wafer is transparent, to access the bulk of Si. This has already led to significant advances[14], in particular with the breakthrough of introducing functionality directly and inside Si, *e.g.*, waveguides, lenses, information

storage and state-of-the-art optical devices[13,15–18]. A recent addition to the emerging applications is the creation of wave plates[19].

While offering unique photonic capabilities, current applications are constrained by the >1 µm fabrication resolution limit[13]. Overcoming this barrier and achieving multi-dimensional nanoscale control inside the wafer would be a major advance. It has the potential to enable 3D nanophotonics[20–22], introduce novel functionalities beyond conventional optics[23,24], and even lead to metasurfaces/metamaterials inside Si. Such devices would have the added benefit of keeping the surface unaltered for parallel micro/nanoscale systems. Most volumetric laser writing relies on Gaussian beams. However, the throughput is generally limited by point-by-point assembly, an inherent limitation of top-down fabrication methods[25]. This was overcome with 3D nonlinear laser lithography, where emergent feedback dynamics[26]

[1]Department of Physics, Bilkent University, Ankara, Turkey. [2]UNAM – National Nanotechnology Research Center and Institute of Materials Science and Nanotechnology, Bilkent University, Ankara, Turkey. ✉e-mail: otokel@bilkent.edu.tr

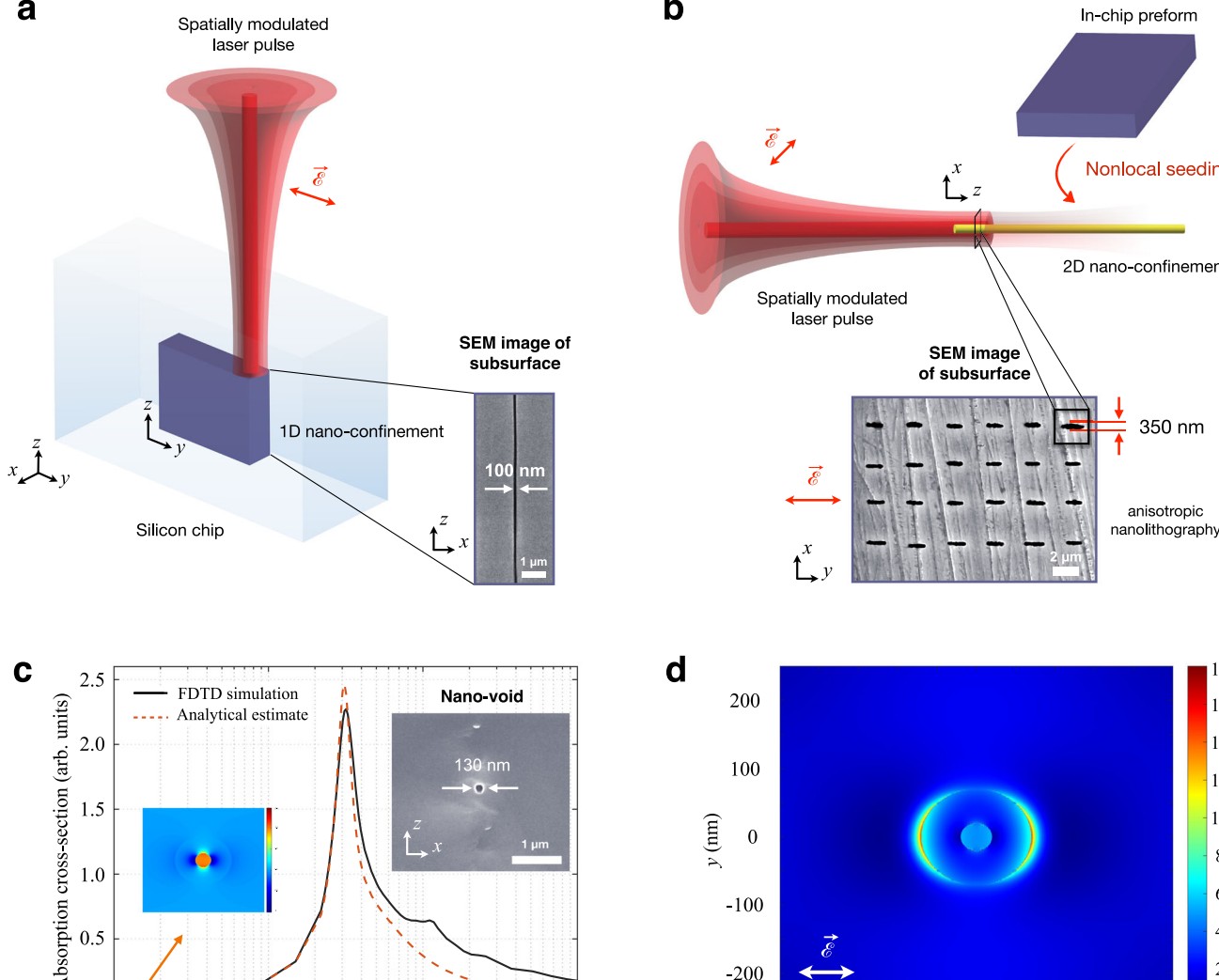

**Fig. 1 | Concept of multi-dimensional laser nanolithography inside Si.**
**a** Spatially-modulated nanosecond laser pulses are used to create non-diffracting beams inside Si. These are exploited to induce planar nanopatterns deep inside silicon (purple-coloured region), without altering the wafer above or below the modifications. Inset: Scanning electron microscope (SEM) image of structures with beyond-diffraction-limit features written inside Si. **b** Using these preformed structures, *i.e.*, in-chip preforms, one can nonlocally seed 2D-confined patterns or nano-lines buried inside Si (yellow region). While the lateral range for seeding is ~ 2 μm, longitudinal writing modality allows the nano-lines to be extended throughout the entire wafer, including to regions where the seed is absent. The nano-lines can be repeated laterally, creating large-area-covering volumetric nanopatterns. Inset: SEM image of the cross-section of large-area covering nano-lines. The alignment and symmetry of nano-lines can be controlled with laser polarisation. **c** 3D finite-difference-time-domain (FDTD) simulations and analytical estimate for field distribution around a 40-nm-diameter nano-void. Inset: SEM image of a representative nano-void in Si. **d** Near-field simulated at the plasmon resonance around a nano-void, enabling polarisation-controlled anisotropic nanolithography. $\vec{\mathscr{E}}$ represents the electric field of the laser.

arising from the interaction of the laser and semiconductor is exploited[13] for rapid bottom-up self-assembly, but with 1 μm resolution[13]. We consider using non-diffractive beams as a promising direction to simultaneously preserve the high throughput and also extend the resolution to the nano-regime. Previous efforts in this direction have not been successful, due to the challenges in balancing competing requirements[14], *i.e.*, low intensity to preserve the wafer surface, and high pulse energy needed to modify the crystal within its bulk[27,28]. Strong beam delocalisation of femtosecond (fs) pulses is a further challenge[27,28].

In this work, we address the fundamental fabrication barrier of achieving controlled laser nanofabrication deep inside the wafer. We use structured beams and exploit anisotropic seeding within the bulk, to create nanostructures of feature sizes down to $100 \pm 20$ nm with subwavelength and multi-dimensional control (Fig. 1). The nanosecond (ns) laser pulses establish control on subsurface patterns based on the details of the spatial beam modulation. The laser polarisation emerges as a further parameter to guide symmetry at the nanoscale. The emerging volumetric nanofabrication capability in turn enables nanophotonics devices inside silicon, namely Bragg gratings, created with multi-level nanofabrication.

## Results

We exploit modulated nanosecond (ns) laser pulses of Bessel type inducing an optical response within Si. The non-diffracting nature of the beam (Supplementary Note 1), along with seeding-based local field enhancement[29,30] (Supplementary Note 2), enables one-dimensional (1D) confinement in the form of nano-planes deep inside Si with

unscathed wafer surfaces (Fig. 1a). The strong energy confinement allows even beyond-diffraction-limit subsurface nanopatterning (Fig. 1a). We also introduce a second laser-writing modality, where local and nonlocal seeding from preformed structures allows 2D-nano-confinement in Si, *i.e.*, nano-lines (Fig. 1b and Supplementary Note 2). The laser polarisation exerts a strong control on the alignment at the nanoscale (inset, Fig. 1b). The basic principle, common to both laser-writing modalities, can be understood based on nonlinear feedback mechanisms within Si (Supplementary Note 2). Initial laser-induced inhomogeneities, *i.e.*, nano-voids (inset, Fig. 1c) result in strong absorption (Fig. 1c). Free carrier density around an inhomogeneity increases due to positive feedback between field intensity and absorption. At around peak absorption, the near-field enhancement (Fig. 1d), and thus additional modifications elongate parallel to the laser polarisation. The updated modification geometry establishes the second feedback mechanism, between morphology and anisotropic field enhancement, thereby enabling polarisation-controlled nanolithography in Si (inset, Fig. 1b and Fig. 3).

We start by discussing the modulated beam profile allowing such unique features (Fig. 2a). We use the Gaussian output of a custom-built fibre laser operating with 10-ns pulses, with up to 9 W power, at the wavelength of $\lambda = 1.55 \, \mu m$ where Si is transparent (see Methods). To generate the required strong energy confinement inside the material, the laser is modulated with a spatial light modulator (SLM), imprinting an axicon phase of zeroth-order Bessel function of the first kind on the beam. The Bessel zone length ($z_B$) and core radius ($d_B$) after SLM are related as, $z_B = w_0 / \tan(\theta)$ and $d_B = 2.4/k \cdot \tan(\theta)$, where $w_0$ is the radius of the incident beam, $k$ is the wave number, and $\theta$ is the cone angle of the virtual axicon. A compromise arises here, where smaller core diameters also correspond to smaller Bessel zone lengths, potentially impeding high-aspect-ratio nanolithography. This can be overcome by implementing a lens-axicon doublet, where strong focusing is invoked by adding an aspheric lens ($L_3$ in Fig. 2a). In this manner, $d_B$ and $z_B$ are decoupled; achieving high-aspect-ratio subsurface nanolithography with powerful scaling and depth control based on laser pulse energy (see Supplementary Note 1).

To generate Bessel beams with SLM, we implement virtual phase profiles $\phi(r)$ of the form,

$$\phi(r) = e^{\pm \left( i2\pi \frac{r}{r_0} \right)}, \tag{1}$$

where $r$ is the radial position and $r_0$ parameter is related to the cone angle $\theta$ as $r_0 = \lambda / \tan(\theta)$. A positive (negative) sign in the complex argument corresponds to a diverging (converging) axicon. The use of a negative sign could seem more intuitive, since it would not require the use of $L_3$, in contrast to the positive case. However, we use the positive phase, as it has the advantage of reduced energy deposition on Si surface, and allows deeper lithography (see Supplementary Note 1). Thus, diverging axicon-lens doublet is used in nanolithography, integrated into a 4-$f$ system for relaying, scaling and spatial filtering (Fig. 2a).

First, we create a Bessel beam profile in air, to be evaluated experimentally and numerically. For optical characterisation, the focused beam is imaged after $L_3$ and its intensity is recorded with 2-μm increments along the optical axis, using an InGaAs camera (Artray, Artcam-031TNIR) coupled to a magnification system. Figure 2b, c show the experimental intensity profiles of a representative Bessel beam ($r_0 = 10$). We represent $r_0$ in units of pixels, *e.g.*, $r_0 = 10$ pixels corresponds to $r_0 = 10 \times 20 \, \mu m$, and to a cone angle of $\theta \sim 7.7$ mrad in the applied phase profile to SLM. The origin of the abscissa in Fig. 2c corresponds to the focal point of $L_3$; followed by the onset of Bessel zone 100 μm later. During lithography, this range is further scaled inside Si due to the high air-Si index contrast. The transverse profile at the maximum intensity point along the optical axis is compared with

simulations (see "Methods") and shown in Fig. 2b along a line passing through the centre of the camera image (inset, Fig. 2b). The comparison confirms the creation of the zeroth-order Bessel function at the output of the SLM-lens doublet in air, which is preserved for a wide range of $r_0$ parameters. Thus, simply by digitally tuning the $r_0$ parameter, modulated beams with diverse energy confinement can be created, analogous to obtaining beam profiles with different physical optics. We exploit such tunable beams of "zeroth-order Bessel function of the first kind" for nanolithography, enabling tunable feature sizes.

To exert control at the nanoscale inside Si, we focus the Bessel beam directly inside the chip, invoking a nonlinear response[13,14] along the non-diffracting, high-intensity Bessel zone. This, in turn induces confined energy deposition and permanent material change. It is important to note that wafer surfaces, as well as bulk crystals above and below lithographic patterns are unaltered after the process. By scanning the sample perpendicular to the laser propagation direction, we create 1D-confined nano-planes (Fig. 2d) in Si, with adjustable thickness along the $x$-axis and strong depth control. The scanning electron microscope (SEM) image of an early laser-written nanopattern (inset, Fig. 2d), indicates 600-nm feature size for an individual nano-plane, and uniform elongation along the $z$-axis of 200 μm. This is achieved with $r_0 = 10$, using a laser pulse of 4-μJ energy, circular polarisation, 1 mm/s scanning speed and single scan per nano-plane (Fig. 2e, left). A detailed analysis of the influence of polarisation and pulse energy will be given later.

Next, we introduce laser lithography of subsurface structures with 2D nano-confinement (*i.e.*, nano-lines). The direct laser writing approach of nano-planes fails to produce any nano-lines. Towards this goal, one can resort to longitudinal scanning (along laser propagation axis) or simply exposing Si to laser without any scanning, however; extensive experiments reveal only micro-lines for a wide range of focusing, laser and SLM parameters (see Supplementary Note 2). We overcome this challenge by invoking a previously unobserved effect in Si, schematically described in Fig. 1b. We first create a buried micro-/nano-plane with transverse writing (Fig. 2e, left). This structure or preform effectively acts as a nonlocal seed to create neighbouring subsurface nano-lines (Fig. 2e, right & Fig. 2f). To reveal these, the sample is polished from the $x$-$y$ plane; followed by treatment with the selective chemical etchant[13]. The existence of the preform is required, as confirmed by Fig. 2g, where nano-lines do not form if the preform is omitted. The seed structure is required to be within ~2 μm distance to induce the formation of a nano-line (see Supplementary Note 2). However, once the nano-line emerges, the structure can be elongated across the entire wafer, simply by scanning the laser along the optical axis (Fig. 2e).

The preceding approach enables the creation of nano-line arrays at all depths, even where the preform is absent. A representative case from a larger array is shown in Fig. 2h at the $x$-$y$ cross-section (the colour-coded subsurface plane is indicated in Fig. 2e, right). The associated preform is created with $r_0 = 10$, while the 300-nm nano-line is created with $r_0 = 6$, pulse energy of 6 μJ, and circular polarisation. The chemical etching is performed for 40 seconds to reveal the cross-section (see Methods). Such structures can be arranged in arbitrary arrays (Figs. 1b, 2f, 3e), potentially covering the entire wafer laterally, limited only by the translational stage range. Thus, the method introduces a powerful nanofabrication capability in Si. A model based on nonlinear thresholding is used to estimate the feature size ($\xi$), guiding our experiments and explaining empirical observations (Supplementary Note 3).

Until now, we focused on the emergence and feature size of nanostructures and constrained the discussion mainly to circular polarisation for maximum symmetry. To further push the limits of nanolithography and to create diverse nano-arrays, we enquire the effect of laser polarisation (Fig. 3). We proceed by inquiring any polarisation effects for 1D-confined structures. We observe that when

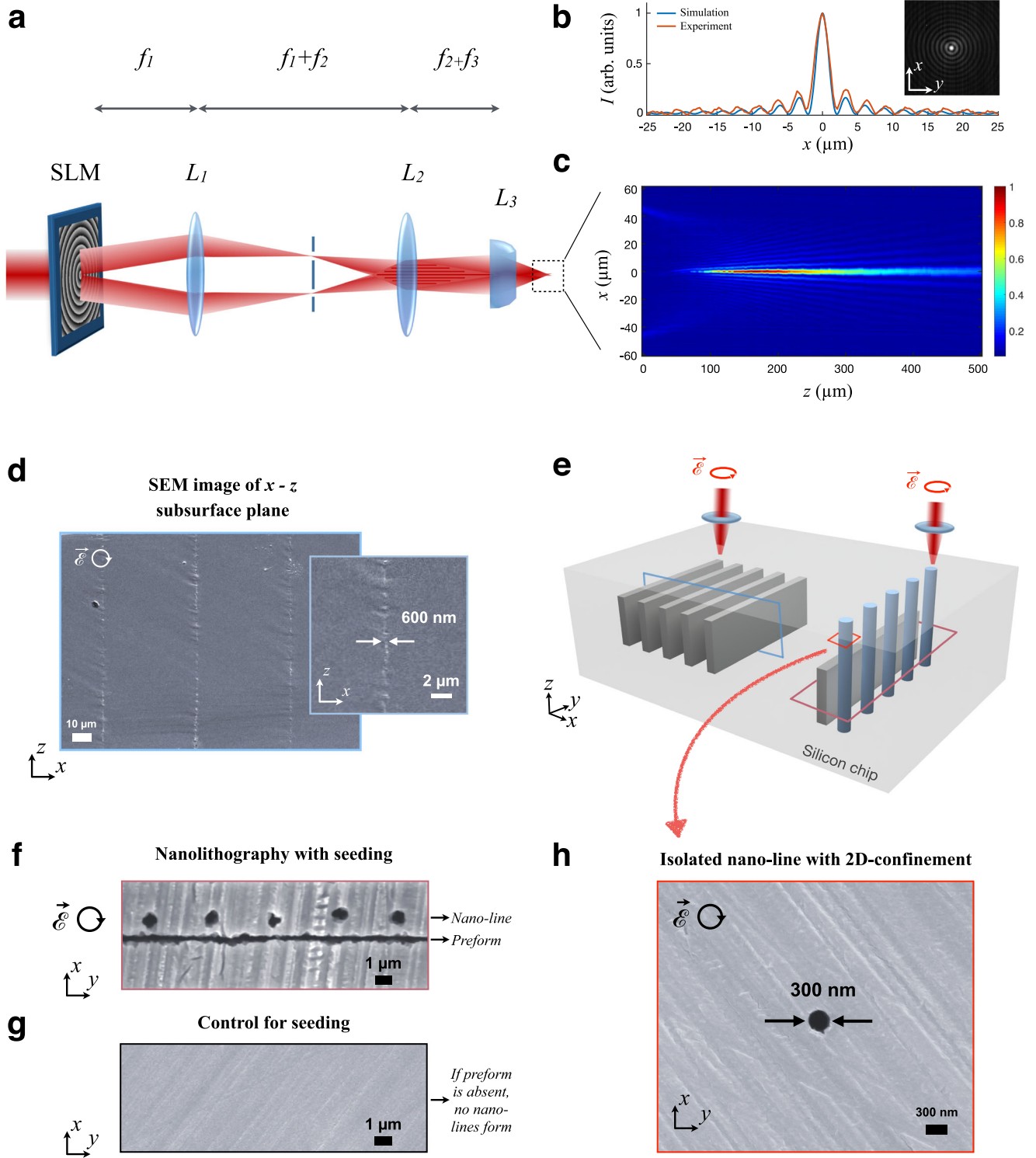

**Fig. 2 | Nanolithography with modulated beams. a** Schematic showing the Bessel beam generated with diverging-axicon and lens doublet. The lenses $L_1$ and $L_2$ serve as a 4-$f$ system that translates the spatial light modulator (SLM) output to the aspheric lens ($L_3$) with a magnification factor of 0.8. **b, c** Beam characterisation in air for $r_0 = 10$. **b** Comparison of simulated and experimental intensity profiles in the transverse plane at maximum intensity plane. Concentric rings are visible in the infrared camera (InGaAs) inset image. **c** Experimental intensity profile recorded on $x$-$z$ plane. In the experimental beam profile, the core diameter is measured as $2\,d_B = 2.15\,\mu m$ and the Bessel zone length as $z_B = 200\,\mu m$, both at full width at half maximum (FWHM). **d** SEM image of the wafer cross-section revealing nano-planes created inside Si. **e** Colour-coded schematic for 1D- and 2D-confined subsurface nano-patterning. The blue cage represents the cross-sectional plane for laser-written structures of (d). The purple cage indicates the cross-sectional plane given in (f). The red cage captures an isolated nano-line indicated in (h). **f** SEM image with a purple border is acquired at a deeper plane, capturing both the seed and nano-line. The seed is created with $r_0 = 10$; the nano-line is created with $r_0 = 6$. **g** The SEM image indicates that nano-lines do not form in the absence of a seed. **h** The SEM image shows an individual nano-line cross-section. 40 seconds etching is used to reveal the structures.

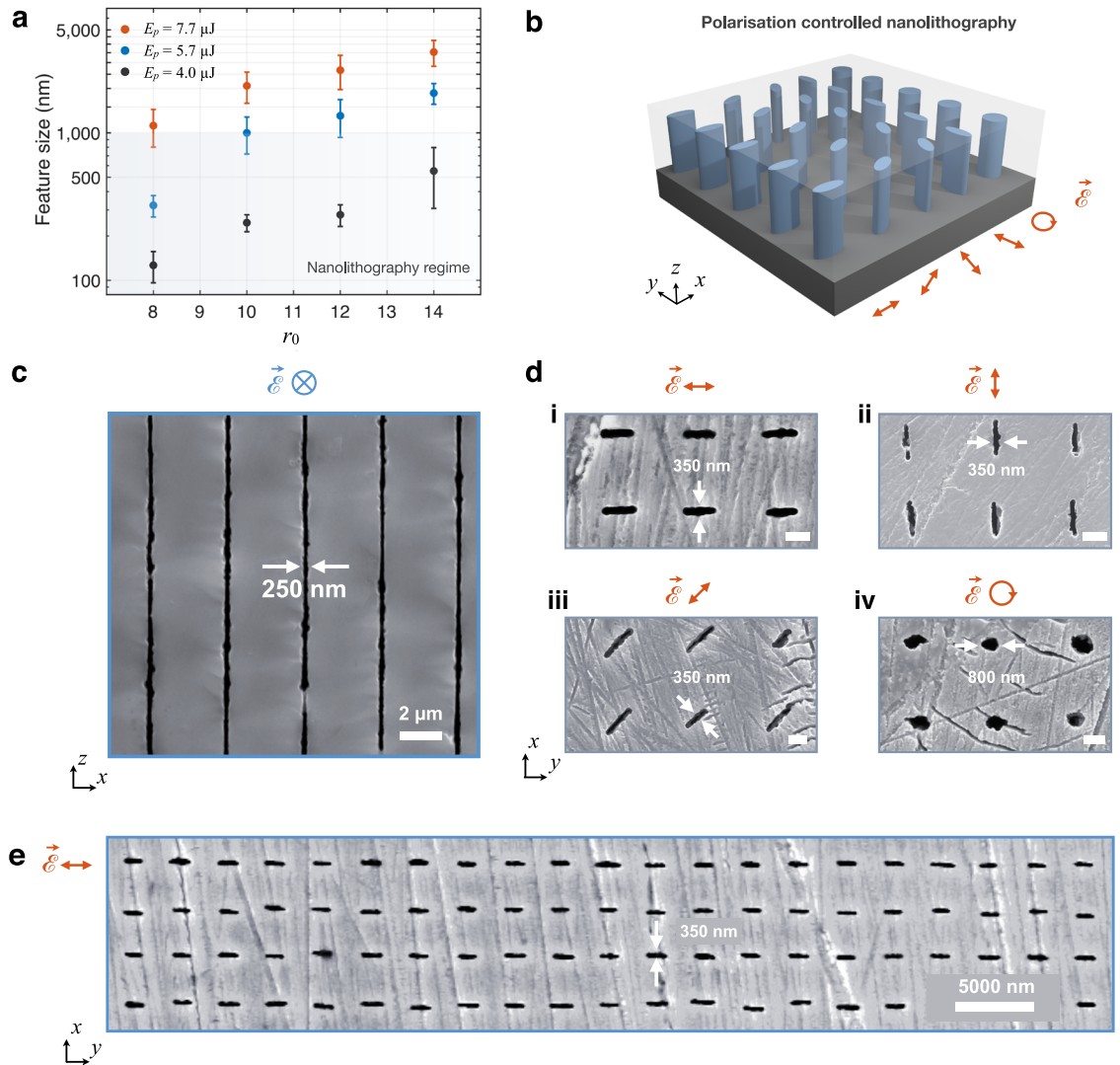

**Fig. 3 | Polarisation-dependent nanofabrication. a** Subsurface nanolithography with 1D confinement, as a function of $r_0$ and $E_p$. Error bars indicate standard deviation. **b** Schematic of polarisation-dependent fabrication with 2D-confinement. Blue sections indicate laser-written nanopatterns with controllable orientation inside Si. **c** SEM image of highly-reproducible, uniform nano-planes created with $r_0 = 10$, $E_p = 4 \mu J$. The structures have a thickness of $250 \pm 30$ nm, at the diffraction limit. The length of the structures along the laser propagation direction is 210 μm. The laser polarisation is parallel to the scanning direction. **d** Architecture of 2D-confined nanopatterns created with various laser polarisations. The thickness along the short axis is 350 nm for linearly-polarised laser writing. The arrays are created with longitudinal-writing modality, exploiting a Bessel beam of $r_0 = 8$ and $E_p = 3.7 \pm 0.3 \mu J$. Scale bar = 1000 nm. **e** SEM image of large-area-covering nano-array created in Si ($r_0 = 8$ with $E_p = 3.7 \mu J$). All samples are polished and etched for 40 seconds to reveal the structures.

polarisation is set to linear and parallel to the scanning direction, the feature size is reduced (see Supplementary Note 4). Using this geometry, systematic experiments were performed for phase ($r_0$) and pulse energy ($E_p$). We find that by simultaneously decreasing $r_0$ and $E_p$, one can reduce the feature size to an order of magnitude smaller than state-of-the-art, down to $126 \pm 30$ nm (Fig. 3a). An SEM image is given in Fig. 3c, showing uniform nanostructuring, and already achieving a diffraction-limited feature size of 250 nm ($r_0 = 10$).

A remarkable phenomenon is observed for in-chip lithography when one considers 2D-confined nanopatterning. At this scale, the symmetry of nanopatterns mimics the orientation of laser polarisation (Fig. 3b). In experiments, laser pulses of $E_p = 3.7 \pm 0.3 \mu J$ propagate along the z-axis; with horizontal, vertical, linear (at 45°) and circular polarisations, with respect to x-axis. We exploit beams with $r_0 = 8$ for creating asymmetric arrays of nano-lines, with planar preforms similar to those given in Fig. 2f. The plethora of nanopatterns created in this manner is shown in Fig. 3d. The feature size in the direction perpendicular to laser polarisation is significantly reduced exploiting linear

polarisation (350 nm), compared to that of circular case (800 nm), which otherwise uses identical fabrication parameters. We achieve unique alignment and orientation control at the nanoscale, scalable with simple scanning (Fig. 3). This behaviour is related to the inherent optical response of the material, which induces nonlinear feedback between morphology and the anisotropic field enhancement (see Supplementary Note 2). Thus, the inherent symmetry of laser polarisation is reflected in the final macro-pattern (Fig. 3b). Such control at the nanoscale also has exciting implications for metamaterial applications within the wafer. A precursor for this direction is large volume coverage, which is experimentally illustrated in Fig. 3e.

Further, by judiciously choosing parameters, such as polarisation, pulse energy, scanning direction, and phase modulation, we achieve a record-low feature size for 1D-confined subsurface nanostructures. This ultimately allows subdiffraction (below 250 nm) lithography (Fig. 4); an exciting new capability implying subwavelength nanophotonics within the bulk. Figure 4a shows a set of experiments with $r_0 = 10$ analysing the critical dimension within the subdiffraction regime; as

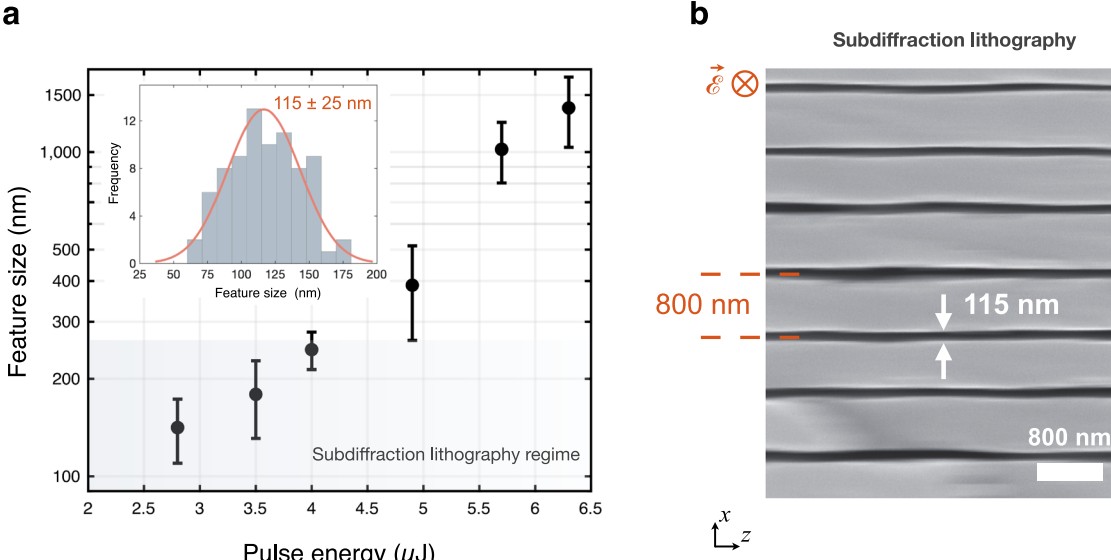

**Fig. 4 | Subdiffraction fabrication regime. a** Plot of mean feature size vs. pulse energy for $r_0 = 10$, illustrating the wide range of control that can be exerted in both nano- and subdiffraction regimes for 1D confinement. Error bars indicate standard deviation. The histogram in the inset shows the size distribution of laser-written structures for $E_p = 3.3 \pm 0.3$ µJ, $r_0 = 10$ from a later data set. **b** The SEM image indicating subdiffraction fabrication with nano-planes with $\xi = 115 \pm 25$ nm and subwavelength modulation of 800 nm, corresponding to the inset in (a). The length of the nanostructures along the $z$-axis is $l = 120$ µm. The laser scanning direction is parallel to the laser polarisation, along the $y$-axis. The structure extent along the $y$-axis is defined by the scanning length; 2 mm in (b). Chemical etching was performed for 40 seconds before SEM imaging.

well as the uniformity of features. Our experiments reveal nanostructures with $115 \pm 25$ nm feature sizes (inset, Fig. 4a) or down to $100 \pm 20$ nm (inset, Fig. 1a, $r_0 = 7$, $E_p = 4.3$ µJ), with optical uniformity. An SEM image from the cross-section of an array of nano-planes with $115 \pm 25$ nm feature size is given in Fig. 4b, indicating nanopatterning capability with sub-micron modulation of 800 nm. These nano-planes have aspect ratios greater than 1000, where the aspect ratio is defined as the ratio of the modification extent along the laser propagation direction ($l$) to that of feature size ($\xi$).

Finally, to showcase the potential of this unique lithographic capability, we create nano-photonics elements deep inside Si (Fig. 5). We start by creating record-efficiency volume-Bragg-gratings (VBG) in Si, towards spectral control inside semiconductors. VBGs are based on refractive-index modulation inside transparent media, patterned over a 3D region[31]. Early Bragg gratings were fabricated in glass, subsequently expanding to other materials and architectures leading to > 90% efficiency and exciting applications[32]. However, analogous components created inside Si are still lacking. To remain within the Bragg regime during 3D fabrication, we use the condition, $l > \frac{n\Lambda^2}{2\pi\lambda}$, where $l$ is the grating length, $n$ is the average refractive index, $\Lambda$ is the grating period and $\lambda$ is the wavelength in air[31]. Operating in transmission mode, incident light is separated into two distinct orders. The highest efficiency $\eta$, i.e., ratio of power in the diffracted beam to that of total transmission, is obtained for incidence at the Bragg angle[33], $\theta_B$, given by $\sin\theta_B = \lambda/2n\Lambda$. Then, using coupled-wave-theory, efficiency for $s$-polarised light can analytically be found as $\eta = \sin^2\left(\pi l \Delta n/\lambda \cdot \cos(\theta_B)\right)$, where $\Delta n$ is laser-induced index contrast[33].

With these considerations, we start fabricating single-layer Bragg gratings with the feature size of $\xi = 700$ nm, period of $\Lambda = 1.5$ µm, and $l = 260$ µm (Fig. 5a). We exploit transverse laser-writing modality, using $r_0 = 7$ and $E_p = 8.7$ µJ (see Methods). A representative VBG operation is given in Fig. 5b, inset. We measured $\eta = 40 \pm 3.5$% for single-layer gratings, which is already higher than the theoretical limit of 33.8% for a thin sinusoidal phase grating[31]. A further approach for improved efficiency $\eta$ is increasing the grating length $l$. Therefore, we introduced multi-level nanofabrication inside Si (see Supplementary Note 5). The level-by-level-written VBGs were created over a large area ($3 \times 3$ mm²)

with 1.6 hour/level speed and then characterised with $\lambda = 1.55$ µm linearly polarised light (see Methods). Figure 5c shows the experimental efficiency values and standard deviations acquired on different positions over the grating area, for different lengths, $l$; along with the corresponding theoretical estimates. We observe excellent agreement between experimental and calculated values, assuming an optical index contrast of $|\Delta n| \cong 1.6 \times 10^{-3}$ between laser-written nanostructures and the crystal matrix. The $\Delta n$ value is independently corroborated with quantitative phase microscopy analysis (see Supplementary Note 6). We achieved the highest efficiency of 87% for double-layer gratings of $l = 490$ µm ($l / \xi = 700$). Further, the angular sensitivity at the Bragg angle is found to be in good agreement with the theoretical estimate (Fig. 5b). The angular bandwidth is measured as $\Delta\theta_{FWHM} = 0.60°$, corresponding to a bandwidth of $\Delta\lambda = 27$ nm.

The preceding results imply that narrow spectral filtering inside Si should be possible (see Supplementary Note 7). To boost angular and, consequently, spectral sensitivity, we fabricated double-level gratings with smaller features, achieving sub-micron modulation. These are of dimensions $\xi = 350$ nm, $\Lambda = 800$ nm and $l = 430$ µm; created with $r_0 = 7$ and $E_p = 6.6$ µJ. The zero-order transmitted beam ($I_0$) is used to characterise the VBG (see Methods). Our experimental results successfully illustrate narrower spectral filtering with control on the central wavelength $\lambda_c$, which are in strong agreement with theoretical predictions (Fig. 5d). For instance, while $\theta_{B2} = 16.10°$ corresponds to a bandwidth of $\Delta\lambda = 10$ nm for $\lambda_c = 1544$ nm; $\theta_{B1} = 15.98°$ results in $\Delta\lambda = 8$ nm with $\lambda_c = 1534$ nm (Fig. 5d). In summary, the achieved features and modulation at the nanoscale ($\Lambda \geq 400$ nm, Supplementary Note 8) enable narrow and tunable spectral response. The presented devices constitute the first nanoscale functional optical elements completely buried in Si.

In conclusion, we report a methodology for creating controlled nanostructures within the bulk of Si, based on 3D nonlinear laser nanolithography, and near-/far-field seeding effects. The nanoscale energy concentration is achieved by the non-diffracting nature of Bessel beams, enabling the creation of nano-voids within the irradiated volumes. This leads to localised field enhancement in their immediate neighbourhood, an effect analogous to the hot spots observed in

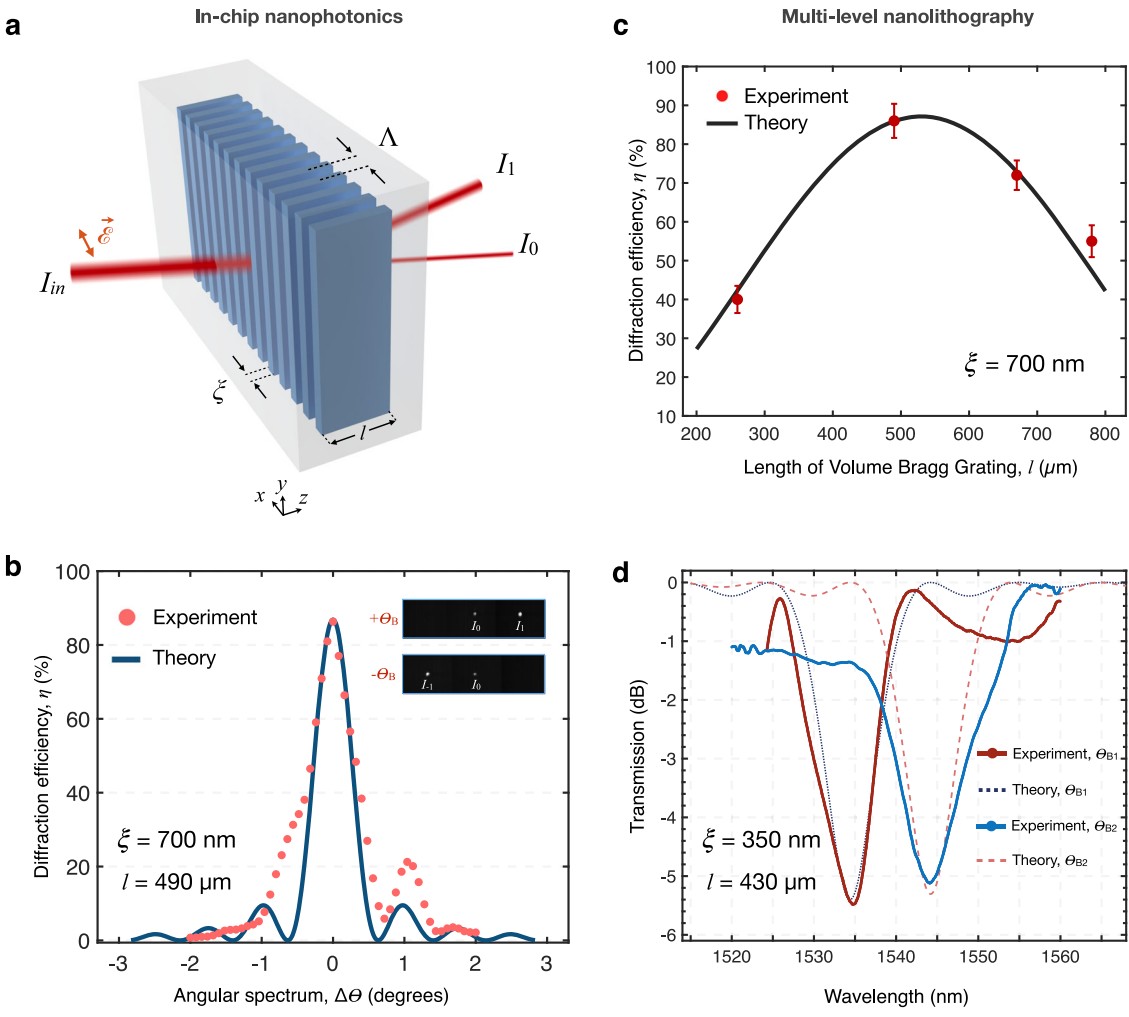

**Fig. 5 | In-chip nanophotonics. a** Schematic of volume Bragg grating (VBG) buried in Si. Nano-planes are created with feature size $\xi$, pitch $\Lambda$, and length $l$, controlled with multi-level nanofabrication. Incident light, $I_{in}$, is separated into two orders over the x-z plane. **b** Angular sensitivity for two-level VBG with $\xi = 700$ nm, $\Lambda = 1.5$ µm, $l = 490$ µm, recorded with s-polarised single-mode laser of $\lambda = 1550$ nm. The angular bandwidth is measured as $\Delta\theta_{FWHM} = 0.60°$, corresponding to a bandwidth of $\Delta\lambda = 27$ nm. The theoretical prediction is given with Kogelnik theory for thick gratings. Inset: Diffraction patterns imaged with InGaAs camera. **c** Diffraction effi-ciencies measured at the Bragg angle for multi-level gratings of $\xi = 700$ nm,

$\Lambda = 1.5$ µm, and various grating lengths, $l$. Data points correspond to one- to four-level writing. The length of single-level nanostructures along the z-axis is 260 µm. Error bars indicate standard deviation. The black curve represents the corre-sponding theoretical calculation. Maximum efficiency of 87% is measured for two-level VBG, $l = 490$ µm. **d** Spectral sensitivity of VBG created with $\xi = 350$ nm, $\Lambda = 800$ nm, and $l = 430$ µm. Spectrum for $I_0$ is measured at the Bragg angle $\theta_{B1} = 15.98°$ resulting in $\Delta\lambda = 8$ nm and $\lambda_c = 1534$ nm; while $\theta_{B2} = 16.10°$ corresponds to a bandwidth $\Delta\lambda = 10$ nm, $\lambda_c = 1544$ nm.

plasmonics but achieved deep inside the wafer. A line roughness in the range of 12–19 nm is evaluated for the nano-planes (Supplementary Note 8), suggesting that the subsurface nanopatterning capability has significant potential for 3D semiconductor nanophotonic devices with low scattering. Further material characterisation studies, including those based on Transmission Electron Microscopy (TEM) and micro-Raman analyses, will contribute to a deeper understanding and use of this capability. The fabricated nanostructures may be expanded to 3D geometries with future advances, potentially through truly 3D holo-graphic laser projection methods[18], by exploiting multi-beam fabrica-tion approaches[34], or through tailoring the nonlinear interactions inherent in the system[35]. We believe the emerging design freedom in arguably the most important technological material will find exciting applications in electronics and photonics; potentially covering the entire near-/mid-IR regime[14]. The beyond-diffraction-limit features (100 nm) and multi-dimensional confinement imply future advances in Si, such as photonic crystals, metasurfaces, metamaterials; numerous information processing applications, with significant potential for integration with on-chip systems[7,36–39]. The introduced nanograting

capability is a step towards this goal, which also constitutes the first multi-layer Si photonics.

## Methods

### Laser nanolithography system and Bessel beam generation

A custom-built all-fibre-integrated master oscillator power amplifier system delivers the pulses for volume nanofabrication. The laser pro-duces pulses in the range of 5–30 ns with a maximum power of 9 W, at a 150 kHz repetition rate. The central wavelength is $\lambda = 1.55$ µm at which Si is transparent. The power and polarisation of the beam are con-trolled by a set of half-wave plates (HWP) and/or quarter-wave plates (QWP). Additional control is achieved using polarising beam splitters (PBS), such as to keep laser polarisation along the orientation of liquid crystal molecules over the SLM surface for optimal efficiency. The beam is expanded with a telescope system ($f = 15$ mm, $f' = 35$ mm), before being reflected from the SLM (Hamamatsu, liquid-crystal-on-silicon SLM X10468-08, 792 × 600 pixels, 20-µm pixel size). The incidence angle on the SLM is kept lower than 10°, with a blazed grating superposed to a phase mask to separate diffraction orders. The

hologram surface is rotated to align the +1 order along the optical axis (z-axis), while other diffraction orders are spatially filtered. The reflected beam is imprinted with the information to create a zeroth-order Bessel beam with an axicon-type phase hologram, and is then imaged onto the focusing lens with a 4-$f$ system ($f_1 = 12.5$ cm, $f_2 = 10$ cm). A magnification factor of 0.8 is used to completely fill the aperture of the final aspheric lens ($f_3 = 4.5$ mm and NA = 0.55), which is used to focus the beam inside silicon for controlled micro/nanofabrication.

For accurate positioning of the beam in Si, the wafer is mounted on a computer-controlled, high-resolution three-dimensional stage (Aerotech, ANT130XY and ANT95-L-Z). For sample preparation, alignment and accurate positioning within the wafer, and selective chemical etching to reveal the structures, previously developed protocols are employed[13]. Experiments were performed at a climate-controlled laboratory ($20 \pm 1$ °C) and in an ambient atmosphere. Depth control is achieved by scanning the laser along the z-axis (Supplementary Note 9). All transverse and longitudinal laser lithography experiments were performed with 1–3 mm/s sample scanning speed.

### Subsurface imaging and material characterisation

Double-side-polished, 1-mm thick, <100 >-cut, p-type (boron-doped, 1–10 Ω.cm) Si samples were used (Siegert Wafer). After laser writing, an in-situ preliminary analysis is performed with a home-built infrared transmission microscope, comprised of a broad-spectrum halogen lamp and a sensitive complementary metal oxide semiconductor (CMOS) camera (Thorlabs, Quantalux, CS2100M-USB). The resolution of in-situ imaging is limited to approximately 1 μm due to the transparency window of Si, thus, for experiments requiring improved resolution, scanning electron microscope (SEM) imaging is employed. To image nano-planes, a surface cut is induced with a diamond-tip cutter, which then is propagated through the crystal revealing the cross-sectional subsurface plane (x-z plane). Analysis of the feature size distributions before and upon $40 \pm 10$ s of etching are observed to be consistent. To image the nano-lines in Si, mainly SEM is employed, due to the challenges in their in-situ analysis. The nano-line images are acquired from the x-y plane upon mechanical polishing, followed by $40 \pm 10$ s selective chemical etching.

### Simulations for Bessel beam profile

To simulate the Bessel beams created in our setup, we employ scalar diffraction theory for monochromatic waves and use Fresnel diffraction expression[31]. First, the desired phase pattern is digitally loaded onto the SLM (792 × 600 pixels). The incident laser is a Gaussian beam of diameter $2w_0 \cong 4$ mm. The spatially-modulated reflected beam is then imaged onto the aperture of the final focusing lens with a 4-$f$ system ($f_1 = 12.5$ cm, $f_2 = 10$ cm), which is then used in nanolithography. The complex field distribution in the observation plane after the SLM is calculated with[31],

$$U_2(r,z) = \left( e^{ikz}/i\lambda z \right) \cdot \mathfrak{I}^{-1}\left\{ \mathfrak{I}\{U_1(r,0)\} \mathfrak{I}\{e^{\frac{i\pi}{\lambda z}r^2}\} \right\}, \quad (2)$$

where z is the distance between the hologram and its projection plane, λ is wavelength, $k = 2\pi/\lambda$ is wave number, r is the polar coordinate with $r^2 = x^2 + y^2$, $U_1(r,0)$ and $U_2(r,z)$ are complex field distributions on hologram and projection planes, respectively. $U_1(r,0)$ is expressed as $U_1(r,0) = U_0 \phi_{SLM}(r)$, where $U_0$ is the incident field amplitude and $\phi_{SLM}(r)$ is the phase pattern digitally loaded to SLM. In our case, different phase distributions are analogous to employing axicons of different cone angles. The demagnification of the system is included in simulations by employing a magnification factor of $f_2/f_1 = 0.8$ for pixel size and beam diameter. Since the 4-$f$ system translates the SLM phase distribution to the final focusing lens ($f_3 = 4.5$ mm and NA = 0.55), the phase profile of lens $\phi_{lens}(r) = \exp(-i\pi r^2/\lambda f)$ is also multiplied with

$U_1(r,0)$. The expression in the diffraction equation then takes the form, $U_1(r,0) = U_0 \phi_{SLM}(r) \phi_{lens}(r)$. In experiments, we used $\phi_{SLM}(r) = \exp(+i2\pi r/r_0)$ corresponding to a diverging axicon, where decreasing the $r_0$ parameter is equivalent to increasing axicon cone angle. The simulations for the field intensity after the focusing lens and with various $r_0$ values are shown in Fig. 2 and Supplementary Note 1, respectively. The simulations, employing 2D Fast Fourier Transform (FFT) algorithm, are performed in MATLAB. Finally, the intensity distribution is calculated from $I_2(x,y) = |U_2(x,y)|^2$.

### Volume Bragg Grating fabrication and optical characterisation

Single-layer gratings were written with transverse laser-writing modality, using a scanning speed of 1 mm/s, repetition rate of 150 kHz, $r_0 = 7$, and $E_p = 6.6$–8.7 μJ. The polarisation of the laser remains parallel to the sample scanning direction during fabrication. After laser writing, VBGs were imaged with an IR transmission microscope verifying their morphology, followed by cross-sectional imaging with SEM for more detailed analysis. Multi-layer VBGs are fabricated with the same set of lithography parameters, however, starting with the deepest layer and consecutively moving closer towards the beam entrance surface for each layer. All gratings were created with a surface area of $3 \times 3$ mm$^2$ in the plane perpendicular to $l$, with 1-mm/s laser scanning speed. Efficiency tests were performed with a diode laser of $\lambda = 1.5$ μm (Thorlabs, FPL1009S), where Si is transparent. The diode laser beam is on the x-z plane and is of s-polarisation (y-axis) before the sample. The diffracted wave is imaged with an indium gallium arsenide (InGaAs, Artray Art-cam-990SWIR) camera. To take into account any losses due to scattering or reflection, the efficiency definition is used as the ratio of power in the +1 diffraction order ($I_{+1}$) to that of total transmitted light ($I_0 + I_{+1}$).

To analyse the spectral response of VBGs, a c-band, p-polarised amplified spontaneous emission (ASE) source covering the spectral range of 1520 nm–1560 nm is used. The zero-order transmitted beam ($I_0$) is coupled to a collimator and directed to an optical spectrum analyser (OSA, Exfo FTBX-5235). The transmission spectrum is then normalised to the spectrum of light passing outside the grating area. The spectral response analysis is repeated for multiple incidence angles. For analytical modelling of efficiency, with its angular and spectral response, we use the general diffraction efficiency equation[33], $\eta = \sin^2(\Phi^2 + \Xi^2)^{1/2}/(1 + \Xi^2/\Phi^2)$, where $\Phi$ is the phase term determining the highest efficiency depending on polarisation and duty cycle, and $\Xi$ is the dephasing parameter which explains deviations from the Bragg condition by detuning $\theta_B$ or $\lambda_c$.

## Data availability

The data that support the results of this paper are available within the manuscript or supplementary information.

## Code availability

The code that supports the results of this paper is available from the corresponding author upon request.

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

## Acknowledgements

This work was supported partially by the TÜBITAK under projects 219M274 (O.T.), 121F387 (O.T.) and by the Turkish Academy of Sciences, TÜBA-GEBIP Award (O.T.). The authors also thank Dr. F. Ömer Ilday for inspiration.

## Author contributions

Experiments were performed by R.A.S., A.I., and A.S. Simulations were performed by R.A.S. and M.B. O.T. arranged the funding, conceived the idea and supervised the project. All authors contributed to the writing of the manuscript.

## Competing interests

R.A.S., A.I., and O.T. are inventors on a patent application (US17/840,023) by the National Nanotechnology Research Center at Bilkent University based on the work presented here. The remaining authors declare no competing interests.
