## [Peer Review File · Nature Communications]

Laser nanofabrication inside silicon with spatial beam modulation and anisotropic seedingREVIEWER COMMENTS

Reviewer #1 (Remarks to the Author):

Building up on their previous work published in Nat. Photonics, the authors present a convincing demonstration of laser-written buried nanostructures inside a Si substrate and this, using nanosecond laser. While writing inside silicon has already been demonstrated (including by the authors), here they achieve smaller feature sizes without introducing any damage on the surface.

The method is based on the use of non-diffractive beams (zeroth-order Bessel beam) to create seeds structure, that are then used to create and propagate nanostructures of various kinds through a field-enhancement due to the presence of the initial seed.

The method of seed-based nanopattern growth has been demonstrated elsewhere (in particular, which should be cited in the main text: <https://doi.org/10.1364/OPTICA.433765> and <https://doi.org/10.1038/s41377-020-0275-2> - the second one is cited in the supplement). Here the novelty is in applying this concept to the bulk-writing of nanostructure buried in silicon, combined with the use of Bessel beams. To the best of our knowledge, this is the first demonstration of this kind in Si.

Throughout the manuscript, the authors demonstrate the process achieving the smallest laser-feature sizes, written in silicon and show how the orientation can be controlled along a plane perpendicular to the optical propagation axis. They also demonstrate how the polarization state can further control the thickness of the nanofeatures and their anisotropic nature. At the end of their study, they demonstrate a volume Bragg grating with an impressive grating efficiency.

The paper is very well written, enhanced with a wealth of details and information in the supplemental materials section. Overall, this is a solid piece of work, complete and meticulously described.

We believe this paper can be instrumental and stimulating further work in this field.

A few points of discussion/improvements:

- Previous work on seed-based writing, albeit not applied to silicon, should be mentioned clearly at the beginning of the manuscript.
- The described method somehow achieves 2D (and 2D and a half) geometries in silicon, due to intrinsic aspect ratio limitations of the Bessel beam. The authors could discuss possible pathways to achieve semi-3D structures or methods that could unlock the third dimension.
- It is not always clear throughout the paper how thick are the structures shown along the propagation axis. (In line 14 of page 4, it is mentioned 200 um?). Just before the authors mentioned a 'strong depth control' (line 12), but what exactly do they mean?
- The statement made in line 19, page 4 is particularly unclear ('Direct application of the preceding approach fails to produce nano-lines'), not sure which preceding approach are we talking about?
- In Figure S2 (b), the arrows pointing out the seed plane seem to indicate two different things on the sketch and the picture. Which one is correct?

Reviewer #2 (Remarks to the Author):

Review Comments on "Laser nanofabrication inside silicon with spatial beam modulation and anisotropic seeding"

In this manuscript, the authors employed spatial beam modulation (Bessel beam) and anisotropic seeding to produce buried nanostructures with subwavelength and multi-dimensional control in silicon. The effects of the phase, pulse energy and laser polarization on nanostructure features have been systematically studied. The nanostructures in silicon hold great potential for applications involving metamaterials. However, this technology is laser-induced structural modification and does not fall within the lithography category. There are numerous studies on the creation of nanostructures within transparent materials, the impact and innovation of this manuscript are lacking. In addition, some issues should be addressed clearly, and mechanisms should be more clearly analyzed.

1. In Page 2, the authors should go into more detail regarding how the research overcomes the difficulty of strong beam decentralization.

2. In Page 2, it cannot be concluded that the effective threshold is reduced through local field enhancement from Fig. 2b. The influence of spatial beam modulation (Bessel beam) on the effective threshold should be further explained.

3. The authors should indicate important laser beam parameters, such as the Bessel zone length (z_B), core diameter (d_B), and cone angle (θ) in Fig. 2a.

4. In Page 4, Line 15, the authors should specify the scanning speed. The scanning speed determines the number of incident pulses per unit area, which affects energy deposition and leads to size changes in the nanostructures.

5. In Page 5, Line 37, the description of the aspect ratio of 2D structures (nano-planes) is incorrect, and the authors should provide a detailed description of the size of the nano-plane. In addition, what factors affect nanopatterning capability with sub-micron modulation?

Reviewer #1

Building up on their previous work published in Nat. Photonics, the authors present a convincing demonstration of laser-written buried nanostructures inside a Si substrate and this, using nanosecond laser. While writing inside silicon has already been demonstrated (including by the authors), here they achieve smaller feature sizes without introducing any damage on the surface.

The method is based on the use of non-diffractive beams (zeroth-order Bessel beam) to create seeds structure, that are then used to create and propagate nanostructures of various kinds through a field-enhancement due to the presence of the initial seed.

The method of seed-based nanopattern growth has been demonstrated elsewhere (in particular, which should be cited in the main text: <https://doi.org/10.1364/OPTICA.433765> and <https://doi.org/10.1038/s41377-020-0275-2> - the second one is cited in the supplement). Here the novelty is in applying this concept to the bulk-writing of nanostructure buried in silicon, combined with the use of Bessel beams. To the best of our knowledge, this is the first demonstration of this kind in Si.

Throughout the manuscript, the authors demonstrate the process achieving the smallest laser-feature sizes, written in silicon and show how the orientation can be controlled along a plane perpendicular to the optical propagation axis. They also demonstrate how the polarization state can further control the thickness of the nanofeatures and their anisotropic nature. At the end of their study, they demonstrate a volume Bragg grating with an impressive grating efficiency.

The paper is very well written, enhanced with a wealth of details and information in the supplemental materials section. Overall, this is a solid piece of work, complete and meticulously described.

We believe this paper can be instrumental and stimulating further work in this field.

We sincerely thank the Reviewer for his/her detailed analysis and enthusiastic comments about the Manuscript. Within the revised version of the Manuscript, we included novel experimental results (*i.e.*, the ground-breaking observation 20-nm sized voids deep inside the wafer), deeper analyses, simulations and extensive experiments (including on roughness, sub-micron modulation, field enhancement simulations and mechanism), and also improved the presentation of the text (please see Suppl. Notes 2, 8 and 9 for additional discussions). When taken together, these refinements significantly enhanced the presentation of the Manuscript, contributing to its overall quality and impact. We also believe that the detailed discussion below will clarify any remaining points.

A few points of discussion/improvements:

- Previous work on seed-based writing, albeit not applied to silicon, should be mentioned clearly at the beginning of the manuscript.

We cite seeding-based writing approaches clearly at the beginning of the Manuscript. Cited articles are, <https://doi.org/10.1364/OPTICA.433765>, and <https://doi.org/10.1038/s41377-020-0275-2>.

The revision is included on Page 2, Lines 30 – 31 as:

"The non-diffracting nature of the beam (Supplementary Note 1, Figure 2b), along with ~~effective threshold reduction~~ seeding-based local field enhancement [28, 29]..."

- The described method somehow achieves 2D (and 2D and a half) geometries in silicon, due to intrinsic aspect ratio limitations of the Bessel beam. The authors could discuss possible pathways to achieve semi-3D structures or methods that could unlock the third dimension.

In our 3D micro-structuring paper we had shown truly 3D fabrication, albeit with feature sizes of 1 μm , and relatively slowly due to point-by-point writing wafer [1]. This had been overcome with multi-beam writing approach, which could potentially help in achieving better dimensional control. Further, one would likely need truly-3D projection capability, which is only recently becoming available [2]. Finally, advanced control on the nonlinear interactions would be required [3, 4]. Each of these would require detailed future research, and relevant discussion is now added in the revision.

The revision is included on Page 7, Lines 11 – 14 as:

"The fabricated nano-structures may be expanded to 3D geometries with future advances, potentially through truly-3D holographic laser projection methods¹⁸, by exploiting multi-beam fabrication approaches³³, or through tailoring the nonlinear interactions inherent in the system³⁴."

- It is not always clear throughout the paper how thick are the structures shown along the propagation axis. (In line 14 of page 4, it is mentioned 200 μm ?).

The elongation along z-direction is related to the Bessel zone length (z_B), which depends on the r_0 parameter (axicon angle) and laser pulse energy, E_p . Figure S1 illustrates the effect of r_0 parameter on the Bessel zone length in air. We add the extent of the nano-structures as:

- In Figure 3c caption, the length along propagation axis is added as 210 μm .
- In Figure 4b caption, the length along propagation axis added as 120 μm .
- In Figure 5c caption, the length of the single-layer grating is added as 260 μm .

- As for the nano-lines, their lengths had been discussed in Page 4, Line 31 of the Manuscript as: **"..once the nano-line emerges, the structure can be elongated across the entire wafer.."**

- Just before the authors mentioned a 'strong depth control' (line 12), but what exactly do they mean?

We thank the Reviewer for the question. One of the important requirements for nanofabrication is depth control. We achieve this by positioning the onset of modifications to the desired depths along the laser propagation direction (z axis). This can be performed by varying the focusing depth in Si. In order to demonstrate this, we created an array of subsurface nano-planes in groups of ten, each group located at a progressively deeper position (Figure 1). In the experiment, the beam propagates along the z axis, where a $\Delta z = +5 \mu\text{m}$ step in air corresponds to a depth change of $n_{\text{Si}} \times \Delta z$ ($\sim 17.5 \mu\text{m}$) inside the wafer. This is confirmed by the measured change in the onset of nanostructures.

Figure 1: Nano-planes fabricated at different depths. Cross-sectional SEM image from a group of nano-planes, which are created at progressively deeper locations within the wafer. Each group of 10 nano-planes are written with 5- μm steps in air, corresponding $n_{\text{Si}} \times \Delta z$ ($\sim 17.5 \mu\text{m}$) deeper steps in Si. In experiments, the onset of the modification for each group is measured to be separated by $\sim 17.5 \mu\text{m}$ as expected. The fabrication parameters are, $r_0 = 7$, $E_p = 7.3 \mu\text{J}$, and $v = 1 \text{ mm/s}$, the laser is raster scanned along the y axis. Sample is briefly etched before imaging. *Inset:* A zoomed SEM image representing the last line of a group, followed by 10 nano-planes of the next group. The observed variability of $\sim 0.3 \mu\text{m}$ on the onset is ascribed to the repeatability of translation stage.

The revision is included as Supplementary Note 9, and referred in Manuscript Page 7, Line 39.

- The statement made in line 19, page 4 is particularly unclear ('Direct application of the preceding approach fails to produce nano-lines'), not sure which preceding approach are we talking about?

Here, in the creation of nano-lines a comparison is made to the direct-laser-writing methodology established for fabricating the nano-planes (Figure 2c, Manuscript). As the nano-planes are created without external seeds or preforms, it would be reasonable to expect the same methodology to apply to the nano-lines. However, as demonstrated in the Figure 2d-ii, Manuscript; the nano-lines do not form without the existence of preforms. Therefore, we introduced the seeding-based nano-line fabrication methodology, as described in the Figure 2c-ii, Manuscript and Figure 2d-i, Manuscript.

This is further clarified in the Manuscript, on Page 4, Line 21 as:

"...Direct **laser writing** approach of **nano-planes** fails to produce **any** nano-lines."

A more detailed discussion is also provided in Supplementary Information, Page 5.

- In Figure S2 (b), the arrows pointing out the seed plane seem to indicate two different things on the sketch and the picture. Which one is correct?

The two arrows point to the seed planes in Figure S2b; both in the sketch and *in-situ* microscope image. However, the infrared microscopy has a limited resolution, constrained by the transmission window of Si, as well as by the shadows of deeper elongated structures. For clarification, we added a zoomed-in and enhanced portion of the image as inset, which can be seen on the upper right corner of Figure S2b. Further clarifications are made in the caption of the image. We also note that the SEM image in Figure 2d-i, Manuscript *directly* shows the nano-plane and nano-lines together.

We revised the Figure S2b as:

We revised the Figure S2 caption as:

"... **Inset: Contrast-enhanced and zoomed area illustrates the preform and six nano-lines together.**"

References

1. O. Tokel, A. Turnalı, G. Makey, P. Elahi, T. Çolakoğlu, E. Ergeçen, Ö. Yavuz, R. Hübner, M. Zolfaghari Borra, I. Pavlov, A. Bek, R. Turan, D. K. Kesim, S. Tozburun, S. Ilday, and F. Ö. Ilday, "In-chip microstructures and photonic devices fabricated by nonlinear laser lithography deep inside silicon," *Nat. Photonics* 11, 639–645 (2017).
2. G. Makey, Ö. Yavuz, D. K. Kesim, A. Turnalı, P. Elahi, S. Ilday, O. Tokel, and F. Ö. Ilday, "Breaking crosstalk limits to dynamic holography using orthogonality of high-dimensional random vectors," *Nat. Photonics* 13, 251–256 (2019).
3. J. Li, J. Yan, L. Jiang, J. Yu, H. Guo, and L. Qu, "Nanoscale multi-beam lithography of photonic crystals with ultrafast laser," *Light Sci. Appl.* 12, 164 (2023).
4. C. Kerse, H. Kalaycıoğlu, P. Elahi, B. Çetin, D. K. Kesim, Ö. Akçaalan, S. Yavaş, M. D. Aşık, B. Öktem, H. Hoogland, R. Holzwarth, and F. Ö. Ilday, "Ablation-cooled material removal with ultrafast bursts of pulses," *Nature* 537, 84–88 (2016).

Reviewer #2

In this manuscript, the authors employed spatial beam modulation (Bessel beam) and anisotropic seeding to produce buried nanostructures with subwavelength and multi-dimensional control in silicon. The effects of the phase, pulse energy and laser polarization on nanostructure features have been systematically studied. The nanostructures in silicon hold great potential for applications involving metamaterials. However, this technology is laser-induced structural modification and does not fall within the lithography category. There are numerous studies on the creation of nanostructures within transparent materials, the impact and innovation of this manuscript are lacking. In addition, some issues should be addressed clearly, and mechanisms should be more clearly analyzed.

We sincerely thank the Reviewer for his/her detailed analysis and enthusiastic comments about the Manuscript. Indeed, upon the Reviewer's input, we included novel experimental results (*i.e.*, the ground-breaking observation 20-nm sized voids created deep inside the wafer), performed deeper analyses, simulations and experiments shedding light on nano-structuring (including on roughness, sub-micron modulation, field enhancement and mechanism), and also improved the presentation of the Manuscript. When taken together, these meticulous refinements significantly enhanced the presentation and clarity of the Manuscript, contributing to its overall quality and impact.

Laser nano-structuring inside insulators, such as polymers and glasses, has been shown, however no such technology exists for nano-structuring inside semiconductors; notably within the technologically critical silicon. Our previous micro-scale lithography work and associated efforts define the state-of-the-art for 3D micro-fabrication inside Si, which have been considered as: "... *With these two schemes, the stealth dicing and some associated technologies have demonstrated high-quality micron-size drilling, dicing or patterning on semiconductors/dielectrics such as through-silicon via (TSV) and through-glass via (TGV), which have revolutionized the microelectronic industry [41-43]*" (Hong-Bo Sun *et al*, <https://arxiv.org/abs/2308.02352>).

Here in contrast, we establish the first large-volume nano-scale laser-writing capability within Si, which is analogous to works in other technologically relevant materials:

- Xu, X. *et al.*, "Femtosecond laser writing of lithium niobate ferroelectric nanodomains", *Nature*, 609, 496, 2022.
- Rodenas, A. *et al.* "Three-dimensional femtosecond laser nanolithography of crystals", *Nat. Photon.*, 13, 105, 2019.
- Wei, D. *et al*, "Experimental demonstration of a three-dimensional lithium niobate nonlinear photonic crystal", *Nat. Photon.*, 12, 596, 2018.
- Xu, T. *et al*, "Three-dimensional nonlinear photonic crystal in ferroelectric barium calcium titanate", *Nature Photon.*, 12, 591, 2018.

Considering silicon's propensity for scaling, its fundamental role for the semiconductor and photovoltaics industries, as well as Si-photonics, one may expect emerging applications similar to the preceding at reduced scales. Using micro-scale fabrication ($> 1\text{-}\mu\text{m}$ resolution) our group and other researchers have already created a plethora of subsurface applications, including waveguides, gratings, lenses, state-of-the-art *Fourier* and *Fresnel* holograms, as reviewed in [1]. In striking contrast, the nano-fabrication capability in this work enables previously impossible functionality to be achieved directly inside the wafer, such as spectral control (Figure 5d, Manuscript).

As such, we achieve the following unique advances.

1. We achieve **$\sim 100\text{-nm}$, beyond-diffraction-limit fabrication resolution** deep inside Si.
2. We achieve **subwavelength modulation** in Si— a critical precursor for 3D meta-materials.
3. We establish the **laser polarisation as a new tool** to control the symmetry at the nanoscale.
4. **Unique seeding-based interactions** inside Si initiate and propagate the nano-structures, constituting unique new physics.
5. We **introduce nanophotonics functionality inside** the wafer. Our proof-of-principle device is unique in its architecture, with potential to open a new research field. As the light wavelength is comparable to feature size, different functionalities emerge.
6. We introduce **multi-level nano-scale fabrication** into the bulk of Si.
7. Plasmonic-based interactions emerge inside the wafer - a new arena for plasmonics.

Notably, none of the preceding can be duplicated with any other technology.

We note that the responsible mechanisms are also unique. We have identified **(i)** near-field-based feedback explaining asymmetric nano-structuring, and **(ii)** far-field effects guiding the fabrication of neighbouring structures. These seeding-type effects are further discussed in the Supplementary Note 2. Critically, with additional experiments we now identify **20-nm-sized nano-voids** buried inside Bessel-beam-written modifications. The associated anisotropic field enhancement and regulatory role in fabrication is discussed in the following questions. However, it is immediately obvious that these nano-voids correspond to the first observation of such structures buried inside Si crystals, with implications in material science, plasmonics and photonics. Such nano-scale control has led to birefringence and nano-gratings in silica glass, which demonstrate excellent multi-dimensional information storage capabilities. The industry is already capitalizing on such capabilities (**Microsoft, Project Silica** and Lei *et al.*, Laser & Photon. Rev. 17, 2200978, 2023).

One may consider two independent strategies to achieve complete 3D nano-lithography within Si. The first is to create controlled networks of 3D-confined nano-voids, which could then be converted into complex devices or Si-photonics. The second is to develop a highly-selective etchant to remove the modified areas towards creating surface/sub-surface nano-patterns with optical behaviour akin to meta-surfaces. Our work here establishes the basis for both strategies.

However, completely buried metamaterials may already be within reach. If one can fabricate nano-structures with sub-micron separation, one may also strongly manipulate the phase of light.

Figure 1. (a) Transmission and (b) Phase output of unit-cell obtained from theory and full-3D FDTD solution. Unit-cell includes buried nano-structures of 300-nm diameter and 534- μm length along propagation direction. 0– 2π phase control is suggested for the period range of 350 - 500 nm.

While such metamaterials are not within the scope of this work, we took it upon ourselves as a challenge and inquired whether completely buried metamaterials would be possible using preliminary simulations. We simulated optical behaviour of subsurface nano-structures assuming sub-wavelength periodicity. The cylindrical nano-structures are of 300-nm diameter and 534- μm length, with an optical index contrast of 9×10^{-3} inside the wafer. The structures formed the unit-cells in an array pattern, with periodicities of $\Lambda = 310 - 500$ nm. The preceding simulation parameters align with the capabilities of our fabrication technique. We then explored periodic nano-modifications with incident light of wavelength 1550 nm. The 3D full-wave simulations are implemented with an FDTD solver, using periodic boundary conditions. The results are summarized in Figure 1. We observe that the phase of the transmitted light depends strongly on the nano-modification separation. Further, complete 0– 2π phase control may be expected for the $\Lambda = 350 - 500$ nm range. The subwavelength unit-cell yields no diffraction order outside the wafer where all the orders are evanescent except the 0th order, and in the 3D FDTD simulation results, we obtain uniform phase and electric field intensity profile. In this condition, one may approximate the system with the semi-theoretical *Fabry-Pérot* transmission function [2] as follows:

$$t_{\text{FP}} = \frac{4(n_{\text{eff}}/n_0)e^{in_{\text{eff}}k_0H}}{(n_{\text{eff}}/n_0 + 1)^2 - (n_{\text{eff}}/n_0 - 1)^2e^{2in_{\text{eff}}k_0H}}$$

, where k_0 is the wavevector in air, n_0 is the optical index of Si, H is the modification length, and n_{eff} is the effective refractive index found in simulation. The results from this equation closely follow the simulation results (Figure 1, theory).

These preliminary simulations indicate the possibility of advanced phase control based on nano-structuring in Si with subwavelength modulation. In principle, one could design pixels of varying phase by changing modulation, rather than controlling the height. While this is only one of the

significant potential directions of laser nano-structuring, we would like to reiterate that metamaterials are not within the immediate scope of this work.

We believe that the list of exciting advances, emerging physics within the bulk of a semiconductor, the nano-scale applications and our preliminary simulations illustrate the impact of the work.

1. In Page 2, the authors should go into more detail regarding how the research overcomes the difficulty of strong beam decentralization.

Strong beam delocalization is associated with complex nonlinear interactions observed with intense femtosecond pulses inside the semiconductor [1, 3, 4], which has so far prevented robust 3D micro-structuring in Si [1]. This has previously been overcome by using nanosecond pulses of Gaussian profile, where nonlinear-feedback-mechanisms initiate and enforce the beam collapse [5]. However, structuring is still limited to micro-scale, both in single- and multi-beam experiments [1]. This is in part due to creating large scattering centres or voids during material modification. If nano-scale voids can be created, these may assist in field enhancement and scattering, as discussed below.

Scaling of voids

Gaussian beam

Bessel beam

Figure 2. The scaling of subsurface void size with the beam modulation. (a, b) Scanning electron microscopy (SEM) images of subsurface voids within modified areas. These voids are created with Gaussian beams and can form at scales of hundreds-of-nanometres. **(c, d)** SEM images subsurface voids within modified areas. These are created with Bessel type beams and are observed to be of few tens-of-nanometres size. All experiments are performed with pulse energy of $E_p = 5.6 \mu\text{J}$, scanning speed of 2 mm/s, and horizontal polarisation. Bessel beam is created with $r_0 = 6$. The SEM data is recorded directly from the wafer cross-sections after cutting; before any etching or post-processing steps on samples. Measured feature sizes are, $\zeta = 3 \mu\text{m}$ in (a,b), and $\zeta = 370 \text{ nm}$ in (c,d).

We performed extensive laser-writing experiments using Gaussian and Bessel beams, in order to evaluate such subsurface voids (Figure 2). Figures 2a-2b show representative voids observed inside modified areas, written with Gaussian beams of $E_p = 5.6 \mu\text{J}$, scanning speed of 2 mm/s, horizontal polarisation, and single-beam writing. While the modified section is of $\zeta = 3 \mu\text{m}$, the voids are observed to be $\sim 300 \text{ nm}$ (Figure 2a), or consist of neighbouring voids of $>200\text{-nm}$ size (Figure 2b).

This observation is in stark contrast compared to the results of Bessel beam experiments (Figures 2c-2d). In these experiments, we used the same laser parameters except the beam modulation, which reduced the feature size to $\zeta = 370 \text{ nm}$. Further, the size of the voids were observed to be on the 20 – 40 nm scale (Figures 2c-2d). *We note that these are the smallest-sized voids so far observed buried inside Si crystals.* The transition from micro-structuring to nano-structuring is found to be self-consistent, in the sense that the void inside the modified area is not larger than the feature size, ζ .

Thus, while the robust energy localization during material modification is sustained due to the remarkable non-diffracting and self-healing characteristics of Bessel beams [6, 7], and the nonlinear self-focusing is achieved due to thermal lensing [5], one may also expect strong field enhancement due to the emergence of nano-scale voids. These also reduce the beam scattering in comparison to the larger-sized voids found in Gaussian-beam experiments (beam scattering scales with the sixth power of the scatterer size in Rayleigh scattering). The nano-voids continually modify and regulate the complex evolution of the system, based on field enhancement. While the complex nonlinear dynamics of feedback-based light-matter interaction is beyond the scope of this work, in order to gain further insight, we performed detailed simulations for the field enhancement, which varies as a function of the nano-void size. The associated simulations are further discussed below.

We now include the preceding discussion in the Supplementary Note 2, Pages 8 – 10. We also revised Page 2, Line 31 as: "The non-diffracting nature of the beam (Supplementary Note 1, Figure 2b), along with ~~effective threshold reduction~~ seeding-based local field enhancement [27, 28] (Supplementary Note 2), enables one-dimensional (1D) confinement at the nanoscale".

2. In Page 2, it cannot be concluded that the effective threshold is reduced through local field enhancement from Fig. 2b. The influence of spatial beam modulation (Bessel beam) on the effective threshold should be further explained.

In order to clarify, we removed the dashed line referring to the threshold-based modification from Fig. 2b, and included the following detailed discussion on effective threshold reduction to Suppl. Note 2. The nonlinear thresholding was used as part of the simplified model in Suppl. Note 3.

"The exact nature of the emergence of nano-voids is currently unknown. However, as they form during material modification, they create strong field enhancement in their immediate environment, analogous to plasmonic nano-particles. We confirm this behaviour with 3D finite-difference-time-domain (FDTD) simulations using LumericalTM solver (Figure 3). We assume nano-voids of varying sizes (2 nm to 300 nm), with a uniform carrier density of $\rho = 3.1 \times 10^{27} \text{ m}^{-3}$ of the same size surrounding voids, corresponding to the peak of absorption in Fig. 1c of the Manuscript. The field enhancement ($|E|^2/|E_0|^2$) is evaluated along a linear cross-section, parallel to the beam polarisation axis. The field is found to be significantly enhanced for small scattering centres, in particular for the sub-40 nm sized voids (Figure 3). This size range matches to the observations of voids in the Bessel beam experiments (Figures 2c-2d). Further, the enhancement is almost entirely absent for larger sized voids, over a few hundred nanometre range (Figure 3). This strongly suggests that as nano-voids emerge at the 20 – 40 nm scale, they would be regulated in size. In contrast, there would be no such regulatory mechanism for the > 250 nm sized voids, in particular for those experimentally observed within the modifications of Gaussian beam experiments (Figures 2a-2b).

Figure 3. Local field enhancement as a function of the void diameter. 3D finite-difference-time-domain (FDTD) simulations are performed for evaluating the field distribution around spherical voids. The simulations are performed with LumericalTM solver, using the Drude model. The void

diameter is scanned from 2 nm to 300 nm, for 40 data points. We assume a uniform carrier density of $\rho = 3.1 \times 10^{27} \text{ m}^{-3}$, corresponding to the peak of absorption in Fig. 1c of Manuscript. The size of this region is the same as the void size. The simulations are performed on an Intel processor (i7-10700F CPU 2.90GHz), over a time period of 180 hours. The mesh size is chosen as 0.2 nm – 5 nm.

Thus, significant field enhancement is expected in Bessel beam experiments, which is considered to effectively reduce the modification threshold during laser writing. Further, the field enhancement is anisotropic as illustrated in Figure 1 of the Manuscript, which also explains the polarisation dependency of the nano-lines morphology observed in our experiments (Figure 3, Manuscript)."

We now include the preceding discussion in the Supplementary Note 2, Pages 10 – 11. We also revised Page 2, Line 31 as: "The non-diffracting nature of the beam (Supplementary Note 1, Figure 2b), along with ~~effective-threshold-reduction~~ seeding-based local field enhancement [28, 29] (Supplementary Note 2), enables one-dimensional (1D) confinement at the nanoscale".

3. The authors should indicate important laser beam parameters, such as the Bessel zone length (z_B), core diameter (d_B), and cone angle (θ) in Fig. 2a.

We add these parameters to Fig. 2a. We add the cone angle on Page 3, Lines 33 – 34, as:

"We represent r_0 in units of pixels, e.g., $r_0 = 10$ pixels corresponds to $r_0 = 10 \times 20 \mu\text{m}$, and to a cone angle of $\theta \sim 7.7 \text{ mrad}$ in the applied phase profile to SLM".

We add remaining parameters in Fig. 2 caption, as: "In the experimental beam profile, core diameter is measured as $d_B = 2.15 \mu\text{m}$ at FWHM, and the Bessel zone length is measured as $z_B = 200 \mu\text{m}$ ".

4. In Page 4, Line 15, the authors should specify the scanning speed. The scanning speed determines the number of incident pulses per unit area, which affects energy deposition and leads to size changes in the nanostructures.

We add the scanning speed on Page 4, Line 16:

"..., circular polarisation, 1 mm/s scanning speed and single scan per nano-plane..."

5. In Page 5, Line 37, the description of the aspect ratio of 2D structures (nano-planes) is incorrect, and the authors should provide a detailed description of the size of the nano-plane.

The structures reported in Page 5, Line 37, are of $\zeta = 115 \pm 25 \text{ nm}$, as measured along the x axis (Figure 4b, Manuscript), and $l \cong 120 \mu\text{m}$ length along the z axis. The aspect ratio is defined as l/ζ , found to be over 1000. The extent along the y axis is a free parameter, determined by the scanning length. We clarify this In Page 6, Lines 1 – 3, and in the caption of Figure 4b, respectively, as:

"The nano-planes have aspect ratio > 1000 , where the aspect ratio is defined as the ratio of the modification extent along the laser propagation direction (l) to that of feature size (ξ)."

" The length of the nanostructures along the z axis is $l = 120 \mu\text{m}$. The laser scanning direction is parallel to the laser polarisation, along y axis. The structure extent along the y axis is defined by the scanning length; 2 mm in (b). "

6. In addition, what factors affect nanopatterning capability with sub-micron modulation?

In order to evaluate nano-patterning capability with sub-micron modulation, we performed extensive experiments, which now allows us to reach a lower modulation level of ($\Lambda = 400 \text{ nm}$, compared to the previous case of $\Lambda = 800 \text{ nm}$ (Figure 4b, Manuscript)).

We conducted a systematic analysis, where nano-planes are fabricated with sub-micron periodicity. The nano-planes are fabricated with a Bessel beam of $r_0 = 7$, $E_p = 5.6 \pm 0.3 \mu\text{J}$, $v = 1 \text{ mm/s}$, and laser polarisation parallel to the sample scanning direction. The translation stage is set to create nano-planes with periodicities (Λ) ranging from 300 nm to 900 nm, followed by sample preparation for SEM analysis (see Methods). The SEM images are then analysed for periodicity and modulation error and the results are summarized in Figure 4. Here, each blue data point corresponds to the mean value that is calculated from a set of 100 measurements; and the associated error bar corresponds to the standard deviation from the same data set. The orange data points show experimental modulation error, which is calculated as the ratio of the standard deviation to the measured mean periodicity.

Figure 4: Experimentally measured periodicity and associated modulation error. Nano-planes are fabricated with different periodicities (Λ), using a Bessel beam of $r_0 = 7$, $E_p = 5.6 \pm 0.3 \mu\text{J}$, $v = 1 \text{ mm/s}$, and laser polarisation parallel to the scanning direction. The periodicity is varied from $\Lambda =$

300 nm to $\Lambda = 900$ nm, while keeping all other parameters the same. The blue data points correspond to the mean value calculated from a set of 100 data points which are extracted from scanning electron microscopy (SEM) analysis, whereas the error bars correspond to the standard deviation calculated for each data set. The orange data points indicate the error in periodicity (modulation error), which are calculated as the ratio of the standard deviation to the measured mean periodicity.

Figure 4 shows that the measured mean value of periodicity closely aligns with the target value. The dashed line, representing a curve of unity slope, visually illustrates this alignment. However, a decrease in periodicity is also accompanied with an increase in the modulation error (Figure 4). We further discuss roughness and reproducibility of these structures below. We first evaluate the measured roughness, followed by assessing the reproducibility at the lowest values, *i.e.*, $\Lambda = 300$ nm.

Roughness analysis: Roughness is a useful metric in nano-fabrication. We use Average Roughness (R_a) definition, which is evaluated with the formula [8]:

$$R_a = \frac{1}{L} \int_0^L |X(z) - X_{avg}| dz ,$$

where X_{avg} is the mean of variable X calculated over a given range, $X(z)$ is the value of the variable for any given z position, and L is the length of the range under evaluation. R_a corresponds to the average deviation from the mean line. The implementation of this equation is illustrated schematically in Figure 5a, where the green curves correspond to the borders of modification, the red curve is the mean value of green curves at each z coordinate, and the orange line denotes the cumulative mean of all x coordinates (X_{avg}). Then, R_a may simply be found as the average of all deviations from this line (the blue line segments). We followed this methodology to calculate R_a values from SEM images of laser-written subsurface nano-planes. A MATLAB routine is written in order to achieve the required edge detection. A representative SEM image used for this approach is shown in Figure 5b, where the green and red curves have the same meaning as in Figure 5a.

Figure 5: Definition and methodology for roughness, R_a . (a) Schematic for roughness calculation. Green curves follow the borders of modified parts, whereas their mean values for each z coordinate form the red curve. The orange line denotes the cumulative mean of all x coordinates (X_{avg}) of the red curve. The blue horizontal lines show deviations from X_{avg} at each coordinate, $(|X(z) - X_{avg}|)$. These are then used in the roughness calculation. (b) A representative SEM image of nano-planes, showing MATLAB edge detection (green curves) and associated mean value for each z (red curve).

Next, we performed a systematic analysis of R_a on nano-planes which are created with periodicities of $\Lambda = 900$ nm, 500 nm, 400 nm, and 300 nm. These structures are fabricated with Bessel beams in transverse writing modality, using $r_0 = 7$, $E_p = 5.6 \pm 0.3$ μJ , $v = 1$ mm/s and linear polarisation. The SEM images after the MATLAB colour coding for edge detection and mean analysis are shown in Figure 6a to 6d, with progressively reduced Λ down to 300 nm. We observe that nano-planes in the $\Lambda = 900$ nm – 400 nm range form with high-quality; $\Lambda = 300$ nm planes are of partly reduced quality.

Figure 6: Roughness analysis. The SEM images are colour-coded with the same methodology described in Figure 5. **(a)** $\Lambda = 900$ nm, **(b)** $\Lambda = 500$ nm, **(c)** $\Lambda = 400$ nm, **(d)** $\Lambda = 300$ nm. The laser writing is performed with Bessel beams in transverse modality, using $r_0 = 7$, $E_p = 5.6 \pm 0.3$ μJ , $v = 1$ mm/s, and linear polarisation.

Using the SEM images in Figure 6, we evaluate the ζ and R_a parameters from > 7500 data points each, as summarized in the Table below.

Table 1. Roughness analysis.

Periodicity, Λ (nm)	900	500	400	300
Feature size, ξ (nm)	210 ± 30	229 ± 34	187 ± 41	145 ± 36
Roughness, R_a (nm)	12	14.3	13.2	14.6

First, we observe that as Λ is reduced the expected periodicity is not affected (Figure 4); however the deviation in ζ tends to slightly increase (Table 1). Second, the R_a value tends to remain within the 12–15 nm range, even down to $\Lambda = 300$ nm (Table 1). Third, in the lowest modulation case of $\Lambda = 300$ nm (Figure 6d), some laser-written areas tend to merge, and nanofabrication uniformity and reproducibility may be reduced. While the non-diffracting nature of Bessel beams and their self-healing property sustain fabrication, such that the nano-planes form with expected periodicities (Figure 4), the merging indicates to a challenge in fabricating high-density nano-planes. This is attributed to beam distortion at very small modulation values, at scales when Λ becomes comparable to the expected feature size, ζ .

Straightness of translation stage: The straightness is defined as the displacement error perpendicular to the principal movement axis. For our stage (Aerotech, ANT130XY, ANT95LZ), the standard deviation of the displacement error is found using the using the stage feedback analysis tool as ~ 8 nm for 1-mm travel range, consistent with the factory datasheet. This value corresponds to less than 1 nm deviation for the preceding cross-sections analyses, too low to contribute to as error in R_a .

The revision is included as Supplementary Note 8, and referred in Manuscript Page 7, Line 6.

References

1. M. Chambonneau, D. Grojo, O. Tokel, F. Ö. Ilday, S. Tzortzakis, and S. Nolte, "In-volume laser direct writing of silicon—challenges and opportunities," *Laser Photon. Rev.* 15, (2021).
2. S. W. D. Lim, M. L. Meretska, and F. Capasso, "A high aspect ratio inverse-designed holey metalens," *Nano Lett.* 21, 8642–8649 (2021).
3. V. V. Kononenko, V. V. Konov, and E. M. Dianov, "Delocalization of femtosecond radiation in silicon," *Opt. Lett.* 37, 3369 (2012).
4. D. Grojo, A. Mouskeftaras, P. Delaporte, and S. Lei, "Limitations to laser machining of silicon using femtosecond micro-Bessel beams in the infrared," *J. Appl. Phys.* 117, (2015).
5. O. Tokel, A. Turnalı, G. Makey, P. Elahi, T. Çolakoğlu, E. Ergeçen, Ö. Yavuz, R. Hübner, M. Zolfaghari Borra, I. Pavlov, A. Bek, R. Turan, D. K. Kesim, S. Tozburun, S. Ilday, and F. Ö. Ilday, "In-chip microstructures and photonic devices fabricated by nonlinear laser lithography deep inside silicon," *Nat. Photonics* 11, 639–645 (2017).
6. F. Courvoisier, R. Stoian, and A. Couairon, "Ultrafast laser micro- and nano-processing with nondiffracting and curved beams," *Opt. Laser Technol.* 80, 125–137 (2016).
7. M. K. Bhuyan, M. Somayaji, A. Mermillod-Blondin, F. Bourquard, J. P. Colombier, and R. Stoian, "Ultrafast laser nanostructuring in bulk silica, a “slow” microexplosion," *Optica* 4, 951 (2017).
8. T. R. Thomas, "Characterization of surface roughness," *Precis. Eng.* 3, 97–104 (1981).

REVIEWER COMMENTS

Reviewer #1 (Remarks to the Author):

I am satisfied with the answers provided by the authors and I would suggest publishing the paper.

Very nice piece of work! Congratulations!

Feedback on reviewer R#2 from reviewer #1

The issues raised by reviewer #2 have been well addressed and with sufficient novel materials that increase the innovative content of the paper. I think, overall, it is an interesting paper suitable for publication.

A small comment is with respect to the Ra measurement (Roughness). I am not sure this is realistic as they only measure over a line and on the edge. A proper measurement would be to have a full surface cross-section and to perform a proper surface analysis (including waviness). I suspect the current data underestimate the true Ra.

Reviewer #3 (Remarks to the Author):

The authors present a novel method for laser nanofabrication of buried nanostructures inside bulk silicon using spatially modulated nanosecond laser pulses and anisotropic seeding effects. This method provides a big advantage over the current state-of-the-art, that has been limited to micro-scale subsurface fabrication in silicon (~ 100 nm, around 10x smaller than previous subsurface laser patterning in silicon). Overall, this study opens exciting new possibilities for 3D laser nanoprocessing of semiconductors.

In the revised manuscript, the addition of new experimental data and analyses had significantly strengthened the impact of the study. In particular, the discovery of extremely small ~ 20 nm voids within the laser-modified zones provides a physical explanation for the nanoscale energy localization.

I have some minor comments for the authors:

You might explain the term "1D-confined nano-planes" when first mentioned. The planes extend in 2D but are confined along one dimension (thickness).

Similarly, clarify the geometry of the "2D confined nano-lines" when first introduced.

The expanded discussion of roughness, uniformity, and reproducibility is quite helpful. But a short comment in the main text on the potential impact of morphological inhomogeneities on optical performance would be nice.

I recommend adding a couple of sentences on the fundamental light-matter interaction mechanism. What specific phenomena allow the extreme nanoscale energy concentration, and how do they overcome thermal diffusion and self-focusing that typically bottleneck precision in laser-semiconductor processing?

A brief mention of future work on TEM, EBSD, or micro-Raman analysis would be helpful.

Minor typos:

Page 1, Line 16: "barriers are two-fold" should be "barriers is two-fold"

Page 2, Line 8: "exploiting infrared lasers where the wafer is transparent" should be "exploiting infrared lasers at wavelengths where the wafer is transparent"

Page 3, Line 83: "where r is radial position" should be "where r is the radial position"

Page 3, Line 34: "The origin of abscissa" should be "The origin of the abscissa"

Page 5, Line 168: "and phase modulation, we achieve record-low feature size" should be "and phase modulation, we achieve a record-low feature size"

Page 7, Line 24: "The central wavelength is $\lambda = 1.55 \mu\text{m}$ where Si is transparent" should be "The central wavelength is $\lambda = 1.55 \mu\text{m}$, at which Si is transparent"

Reviewer #1

I am satisfied with the answers provided by the authors and I would suggest publishing the paper.

Very nice piece of work! Congratulations!

The issues raised by reviewer #2 have been well addressed and with sufficient novel materials that increase the innovative content of the paper. I think, overall, it is an interesting paper suitable for publication.

We sincerely thank the Reviewer for his/her enthusiastic comments and the recommendation for publication. We included a few minor revisions given as below.

A small comment is with respect to the R_a measurement (Roughness). I am not sure this is realistic as they only measure over a line and on the edge. A proper measurement would be to have a full surface cross-section and to perform a proper surface analysis (including waviness). I suspect the current data underestimate the true R_a .

Currently it is not possible to cut a laser-modified volumetric nano-plane inside Si over its own plane, without introducing additional roughness. To gain further insight into the waviness of such a nano-surface, a useful proxy is to evaluate R_a over two orthogonal cross-sections. We performed R_a analysis over two such perpendicular cuts ($x - z$ plane and $x - y$ plane), and found similar feature size, standard deviation and roughness values over these cross-sectional planes. This analysis is added to the Supplementary Note 8, as:

"Roughness analysis over orthogonal cross-sections: Currently, it is not possible to cut a laser-modified volumetric nano-plane over its own plane at nanoscale, without introducing additional roughness. Thus, it would not be practical to directly evaluate R_a over a laser-written nano-surface. In order to gain further insight into the waviness of a buried nano-plane, a useful proxy is to evaluate R_a over two orthogonal cross-sections. We performed R_a analysis over two such perpendicular planes ($x - z$ plane and $x - y$ plane), which include the laser propagation direction (z direction) and the sample scanning direction (y direction), respectively. We fabricated subsurface nano-planes using $r_0 = 6$, $E_p = 6.6 \pm 0.3 \mu\text{J}$, $\lambda = 800 \text{ nm}$, $v = 1 \text{ mm/s}$ using linear polarisation parallel to the scanning direction. We then polished the sample ($\sim 150 \mu\text{m}$) to reach the $x - y$ cross-section of the nano-structures, followed by an orthogonal cut to reach their $x - z$ cross-section. Brief etching is applied to reveal the nano-structures. For the nano-structures over the $x - z$ plane, we evaluated the feature size as $\xi = 142 \pm 33 \text{ nm}$ and the roughness as $R_a = 19 \text{ nm}$. For the nano-structures over the $x - y$ plane, we found $\xi = 168 \pm 31 \text{ nm}$, with a roughness of $R_a = 18 \text{ nm}$. Thus, similar feature size, standard deviation and roughness are found for the nano-planes over two orthogonal cross-sections."

Reviewer #3

The authors present a novel method for laser nanofabrication of buried nanostructures inside bulk silicon using spatially modulated nanosecond laser pulses and anisotropic seeding effects. This method provides a big advantage over the current state-of-the-art, that has been limited to micro-scale subsurface fabrication in silicon (~100 nm, around 10x smaller than previous subsurface laser patterning in silicon). Overall, this study opens exciting new possibilities for 3D laser nanoprocessing of semiconductors.

In the revised manuscript, the addition of new experimental data and analyses had significantly strengthened the impact of the study. In particular, the discovery of extremely small ~20 nm voids within the laser-modified zones provides a physical explanation for the nanoscale energy localization.

We sincerely thank the Reviewer for his/her detailed analysis and enthusiastic comments about the Manuscript, and the emphasis on the strength of our study, as well as its potential to open exciting new possibilities for 3D laser nano-processing of semiconductors. Indeed, the discovery ~20 nm voids within the laser-irradiated zones provides a physical explanation of energy localisation.

1. I have some minor comments for the authors:

You might explain the term "1D-confined nano-planes" when first mentioned. The planes extend in 2D but are confined along one dimension (thickness). Similarly, clarify the geometry of the "2D confined nano-lines" when first introduced.

Done. We added the former as follows, in Page 2, Line 31, Manuscript.

"... enables one-dimensional (1D) confinement in the form of nano-planes .."

This was also repeated in Page 4, Line 12. The latter had been given in its first mention, Page 2, Line 35 as: "... allows 2D-nano-confinement in Si, *i.e.*, nano-lines ..."; also in Page 4, Line 20.

2. The expanded discussion of roughness, uniformity, and reproducibility is quite helpful. But a short comment in the main text on the potential impact of morphological inhomogeneities on optical performance would be nice.

We added a short comment in Page 7, Line 15, Manuscript, as:

"A line roughness in the range of 12-19 nm is evaluated for the nano-planes (Supplementary Note 8), suggesting that the subsurface nanopatterning capability has significant potential for 3D semiconductor nanophotonic devices with low scattering."

3. I recommend adding a couple of sentences on the fundamental light-matter interaction mechanism. What specific phenomena allow the extreme nanoscale energy concentration, and how do they overcome thermal diffusion and self-focusing that typically bottleneck precision in laser-semiconductor processing?

The extreme nanoscale energy concentration is enabled by the non-diffracting nature of the Bessel beam and the creation of ~20-nm sized voids. Currently, the details of nano-void formation is not known, which will be the topic of exciting future research. However, the formation of nano-voids leads to localised field enhancement in their immediate proximity, enabling an effect analogous to the hot-spots observed in plasmonics, but notably achieved deep inside the wafer. These points are summarised in Page 7, Line 11, Manuscript:

"The nanoscale energy concentration is achieved by the non-diffracting nature of Bessel beams, enabling the creation of nano-voids within the irradiated volumes. This leads to localised field enhancement in their immediate neighbourhood, an effect analogous to the hot-spots observed in plasmonics but achieved deep inside the wafer".

4. A brief mention of future work on TEM, EBSD, or micro-Raman analysis would be helpful.

We add the following comment on Page 7, Line 17, Manuscript:

"Further material characterisation studies, including those based on Transmission Electron Microscopy (TEM) and micro-Raman analyses will contribute to a deeper understanding and use of this capability".

5. Minor typos:

Page 1, Line 16: "barriers are two-fold" should be "barriers is two-fold"

Page 2, Line 8: "exploiting infrared lasers where the wafer is transparent" should be "exploiting infrared lasers at wavelengths where the wafer is transparent"

Page 3, Line 83: "where r is radial position" should be "where r is the radial position"

Page 3, Line 34: "The origin of abscissa" should be "The origin of the abscissa"

Page 5, Line 168: "and phase modulation, we achieve record-low feature size" should be "and phase modulation, we achieve a record-low feature size"

Page 7, Line 24: "The central wavelength is $\lambda = 1.55 \mu\text{m}$ where Si is transparent" should be "The central wavelength is $\lambda = 1.55 \mu\text{m}$, at which Si is transparent"

All are revised as suggested.

REVIEWERS' COMMENTS

Reviewer #1 (Remarks to the Author):

I am satisfied with the answers provided to the reviewers comments and recommend accepting the paper.

Reviewer #3 (Remarks to the Author):

Excellent work. Thank you.